# Near-perfect precise on-target editing of human hematopoietic stem and progenitor cells

**Fanny-Mei Cloarec-Ung[1], Jamie Beaulieu[1], Arunan Suthananthan[1], Bernhard Lehnertz[1], Guy Sauvageau[1], Hilary M Sheppard[1,2], David JHF Knapp[1,3]\***

[1]Institut de Recherche en Immunologie et en Cancérologie, Université de Montréal, Montéal, Canada; [2]School of Biological Sciences, Faculty of Science, University of Auckland, Auckland, New Zealand; [3]Département de Pathologie et Biologie Cellulaire, Université de Montréal, Montréal, Canada

**Abstract** Precision gene editing in primary hematopoietic stem and progenitor cells (HSPCs) would facilitate both curative treatments for monogenic disorders as well as disease modelling. Precise efficiencies even with the CRISPR/Cas system, however, remain limited. Through an optimization of guide RNA delivery, donor design, and additives, we have now obtained mean precise editing efficiencies >90% on primary cord blood HSCPs with minimal toxicity and without observed off-target editing. The main protocol modifications needed to achieve such high efficiencies were the addition of the DNA-PK inhibitor AZD7648, and the inclusion of spacer-breaking silent mutations in the donor in addition to mutations disrupting the PAM sequence. Critically, editing was even across the progenitor hierarchy, did not substantially distort the hierarchy or affect lineage outputs in colony-forming cell assays or the frequency of high self-renewal potential long-term culture initiating cells. As modelling of many diseases requires heterozygosity, we also demonstrated that the overall editing and zygosity can be tuned by adding in defined mixtures of mutant and wild-type donors. With these optimizations, editing at near-perfect efficiency can now be accomplished directly in human HSPCs. This will open new avenues in both therapeutic strategies and disease modelling.

**\*For correspondence:**
david.knapp@umontreal.ca

## eLife assessment

This study presents an **important** methodology to increase the efficiency and precision of gene editing in human hematopoietic stem and progenitor cells. The evidence supporting the claims is **convincing** in that primitive LTC-ICs were minimally affected as a result of the editing procedure and the lack of edits at predicted off-target sites. The work will be of interest to biologists studying hematopoietic stem and progenitor cells and genome editing for potential clinical applications.

## Introduction

Precise genome editing holds substantial promise for both accurate disease modelling and for potential curative treatments for monogenic disorders. There has been a particular interest in therapeutic editing in the hematopoietic system due to the ability to transplant cells to give life-long grafts, and the relatively large number of monogenic disorders that could be treated by gene repair/replacement. Precision edits can be generated using a CRISPR/Cas system to introduce a break at the target locus together with the addition of a template DNA to engage the homology-directed repair (HDR) pathway and insert the edit of interest (*Adli, 2018*). These templates contain the change of interest flanked by homology arms matching the sequence on either side of the break site. While this strategy

can function, non-homologous end joining (NHEJ) which results in random insertions and deletions (indels) is the default pathway, as HDR can only operate in the S and G2 phase of the cell cycle (*Lin et al., 2014*; *Heyer et al., 2010*). As such, the efficiency of HDR-mediated editing, particularly in relevant primary human stem and progenitor populations remains limited, with most strategies achieving efficiencies in the range of 10–20% (*Bak et al., 2018*; *Charlesworth et al., 2018*). Despite such limited efficiencies, multiple clinical trials using CRISPR/Cas-based editing strategies are currently ongoing, reinforcing the extreme interest in the area (*Porteus et al., 2022*).

One simple method to improve HDR efficiency involves increasing the concentration of donor template as intracellular donor concentration has been shown to correlate to HDR efficiency (*Lee et al., 2017*). Indeed, in HSPCs high concentrations of donor AAV have been shown to improve the efficiency of HDR, though likely at the cost of cell viability as, while not measured directly, a subsequent live-dead selection was performed in the study (*Tran et al., 2022*). As cell numbers are limiting and functionality is critical, editing toxicity is as important as raw efficiency. To this end, transient knockdown of p53 by siRNA was shown to improve survival of human hematopoietic stem cells (HSC), though not directly modulate the HDR editing efficiency (*Lehnertz et al., 2021*). Interestingly, Inhibition of 53BP1 which has the additional effect of NHEJ inhibition has been shown to improve the editing efficiency in these cells reaching up to ~60% (*De Ravin et al., 2021*). Critically, both studies demonstrate that edited cells retained their functional stem cell capacity (long-term engraftment in xenotransplantation) following the editing protocol, with efficiencies similar to pre-transplant measurements. Another small molecule inhibitor of NHEJ, M3814 that inhibits DNA-PK, has been shown to increase editing efficiencies up to 80–90% in primary human T cells (*Shy et al., 2023*). M3814 also has the benefit of being a small molecule rather than protein like the inhibitor of 53BP1, thus making it cheaper and easier to use. Importantly for safety, the 53BP1 study also showed that while NHEJ inhibition increased the rate of large deletions in the absence of HDR donor, when HDR donor was present the rate of large deletions was not increased, nor was the rate of off-target edits (*De Ravin et al., 2021*). Inhibition of NHEJ is thus likely a safe and effective way to improve the rates of HDR.

In this study, we optimized ribonuclear protein (RNP) concentration and selection, donor type and design, and small molecule additives, and combined this with optimal culture time and conditions needed to induce division while maintaining stemness in primary human HSC (*Knapp et al., 2017*; *Fares et al., 2014*) in order to further refine HDR in these cells. We demonstrate that while M3814 increases HDR editing efficiency in human HSPCs, a more specific DNA-PK inhibitor AZD7648 (*Fok et al., 2019*) is capable of further improving this efficiency. Combining AZD7648 with optimal pre-stimulation, RNP, p53 siRNA, and AAV donor concentrations yielded mean efficiencies of 97% editing with minimal toxicities. Surprisingly, these conditions worked at equivalent efficiencies for short single-stranded oligodeoxynucleotide donors (ssODNs) giving mean efficiencies of 94% when the ssODNs were modified to mutate not just the PAM sequence but also multiple positions in the spacer (silent mutations). These donors which thus far have not previously been effective in HSPC are much faster and easier to design and are fully synthetic, thus facilitating rapid prototyping and downstream translation. Importantly, the editing protocol is consistent across the hematopoietic hierarchy, and does not affect lineage choice in CFC assays or high self-renewal potential long-term culture-initiating cell frequency. To further facilitate disease modelling applications where zygosity is an important consideration, we also demonstrated that zygosity can be tuned by providing a mixture of silent and mutant donors. Our refined protocol will thus facilitate both therapeutic editing strategies and disease modelling strategies.

## Results

### Precise editing can be achieved using both AAV and ssODN donors in cord blood CD34+ cells

As a basis for our protocol (*Figure 1A*), we used a 48 hr pre-stimulation in a growth factor mixture of SCF, FLT3L, IL3, and IL6 which we have previously shown induces even the most primitive phenotypic HSC to cycle by ~66 hr (*Knapp et al., 2017*), thus ensuring that editing machinery would be present during S/G2 phase of the cells. We also included UM171, which improves the maintenance of HSC functional capacity (*Fares et al., 2014*) and can improve transduction efficiency in these cells (*Ngom*

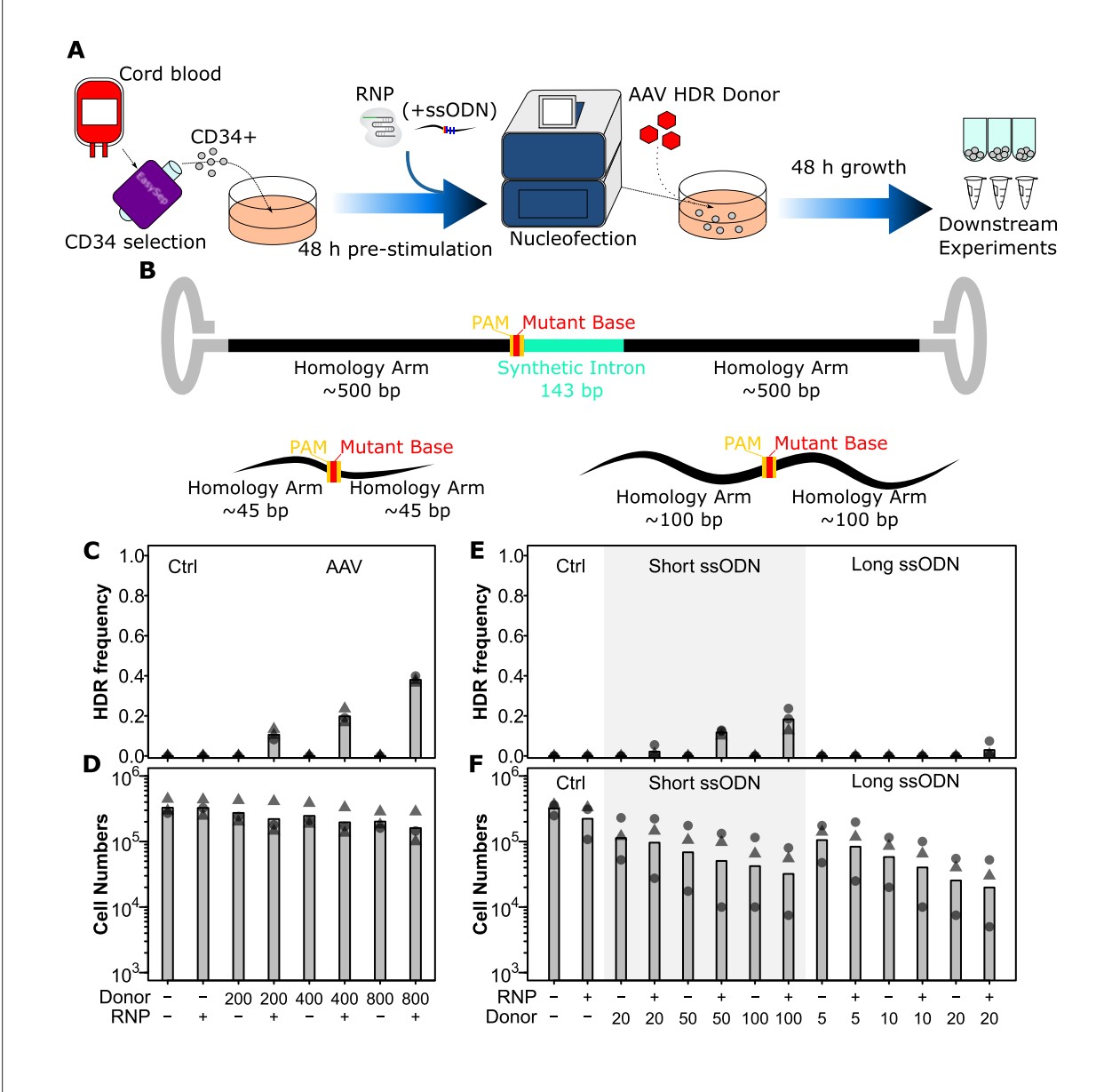

**Figure 1.** AAV and short ssODN both allow precise editing in human hematopoietic stem and progenitor cells (HSPCs). (**A**) Experimental design. (**B**) Homology-directed repair (HDR) donor configurations. The SRSF2 P95H AAV donor is shown above and short and long ssODN donors are shown below with features indicated. Annotated sequences are shown in *Supplementary file 2*. (**C**) *HDR integration efficiency by AAV dose.* Cells were edited with 30.5 pmol ribonuclear protein (RNP) (or not as indicated) with indicated multiplicities of infection (MOI) of AAV donor. Bars show mean values and points show measurements for individual cords. Male cords are shown as triangles and females as circles. (**D**) *Viable cell number by AAV dose.* Hemocytometer counts at the time of harvest are shown for each sample from (**C**). (**E**) *HDR integration efficiency for short and long ssODN donors.* Donor DNA amounts are shown in pmol. (**F**) *Viable cell number by ssODN dose.* False-discovery rate (FDR) corrected paired t-test significance values are shown in *Supplementary file 1*. See also *Figure 1—figure supplements 1 and 2*.

The online version of this article includes the following figure supplement(s) for figure 1:

**Figure supplement 1.** Ribonuclear protein (RNP) efficiency determination.

**Figure supplement 2.** Assays for the detection of homology-directed repair (HDR) integration.

*et al., 2018*). These conditions are similar to other reported editing strategies (*Bak et al., 2018*). As a first step in our optimization, we tested different conditions for gRNA-mediated cutting. We tested two concentrations of RNP with and without an electroporation enhancer (IDT) on a site in the SRSF2 locus (*Figure 1—figure supplement 1*). Cutting efficiency was assessed 48 hr post electroporation

using the T7E1 assay (*Guschin et al., 2010*) and toxicity by counting the absolute cell number for each condition on the same day (*Figure 1—figure supplement 1*). At lower RNP concentrations, we observed a slight benefit of the addition of an electroporation enhancer, though this was not present at higher concentrations (FDR = 0.02 and FDR = 0.48, respectively, *Figure 1—figure supplement 1A*). Similarly, increasing RNP concentration also showed a benefit to overall editing efficiency (FDR = 0.08, *Figure 1—figure supplement 1A*). Both the addition of an electroporation enhancer and the increase in RNP concentration, however, resulted in significantly decreased cell numbers indicating toxicity (*Figure 1—figure supplement 1B*). Similar patterns were observed at the SF3B1 locus, though overall efficiency was lower for this gRNA (*Figure 1—figure supplement 1C and D*). As the improvements in cutting efficiency, while statistically significant, were only in the range of ~10% and both increased toxicity, we selected 30.5 pmol RNP without enhancer for subsequent experiments.

We next tested three different types of donors: AAV, short ssODN, and long ssODN (*Figure 1B*). Both ssODN donors were co-delivered in the same nucleofection as the RNP, while the AAV donors were delivered within 15 min of the electroporation as reported by *Charlesworth et al., 2018*. Integration was assessed 48 hr later by PCR and gel quantification. For the AAV donor, this could be done by assessing product size directly as it included an ~100 bp synthetic intron. One critical factor we discovered here was that for this assessment a nested PCR was required with outer primers outside the homology arms of the AAV, as otherwise non-specific amplification from non-integrated donors predominated (*Figure 1—figure supplement 2A*). For ssODN donors, a BspEI site was introduced, allowing assessment based on an enzymatic digest. Using defined mixtures of WT and BspEI DNA, we show that this assay is accurate down to ~1% integration efficiency (*Figure 1—figure supplement 2B*). As with RNP, we also assessed cell toxicity by absolute counts at the time of harvest. For AAV, a multiplicity of infection (MOI) of 400 MOI gave the highest integration rate prior to observable toxicity with an integration rate of 19.8% (*Figure 1C and D*). This integration is consistent with what can be found in the literature for this type of donor (*Bak et al., 2018*; *Charlesworth et al., 2018*; *Lehnertz et al., 2021*). While the long ssODN donors showed a very low integration efficiency of 3% at 20 pmol and a high toxicity, short ssODN donors were more promising with an integration rate of 12% at 50 pmol and a lesser though still appreciable toxicity (*Figure 1E and F*). Higher concentrations of short ssODN further raised efficiency though with a corresponding increase in toxicity beyond a usable range (*Figure 1E and F*). Interestingly, toxicity for both ssODN donors was independent of the presence of the Cas9 nuclease (equivalent in the no RNP condition), suggesting that it is likely an innate immune reaction to the single-stranded DNA rather than the editing process itself. These data demonstrate that the AAV donor at 400 MOI and short ssODN at 50 pmol are the most suitable donors.

## Optimal inhibition of NHEJ enables integration efficiencies up to 100% with both AAV and ssODN donors

To further increase HDR efficiency, we next tested whether the DNA-PK inhibitor M3814 could also boost HDR efficiency in HSPCs. In these tests, AAV donors were used and p53 siRNA included to maximize survival. We observed an increased HDR efficiency up to 70% with M3814 (*Figure 2A*), equivalent to reports with inhibition of 53BP1 (*De Ravin et al., 2021*). We also tested another DNA-PK inhibitor AZD7648 which has been reported to inhibit DNA-PK with reduced off-target activity on PI3K (*Fok et al., 2019*). Interestingly, we observed improved editing efficiency at 0.5 µM with AZD7648 compared to M3814, and 0.5 µM AZD7648 was equivalent to 5 µM M3814 (*Figure 2A*, FDR = 0.01 and FDR = 0.24, respectively). Toxicity, as assessed by viable cell counts, was also marginally but statistically significantly lower for AZD7648 compared to M3814 at 0.5 µM (*Figure 2B*, FDR = 0.02 for 0.5 µM). There was also no significant difference in editing efficiency for AZD7648 at 0.5 and 5 µM but a slight improvement in viability (FDR = 0.16 and FDR = 0.01, respectively *Figure 2A and B*). We next tested whether the addition of RS1, a RAD51 stabilizer and thus HDR enhancer could further improve efficiencies (*Song et al., 2016*). In this context, the addition of RS1 either alone or together with AZD7648 was detrimental to editing efficiency and toxic to cells (*Figure 2C*). The combination of p53siRNA and AZD7648, however, was able to achieve mean efficiencies of 80% and up to 96% in these tests with minimal toxicity (*Figure 2C and D*). Repeating these tests on the SF3B1 locus revealed similar trends, though with a somewhat lower mean editing efficiency of 57% (*Figure 2— figure supplement 1A and B*). This difference reflects the lower cutting efficiency observed at this

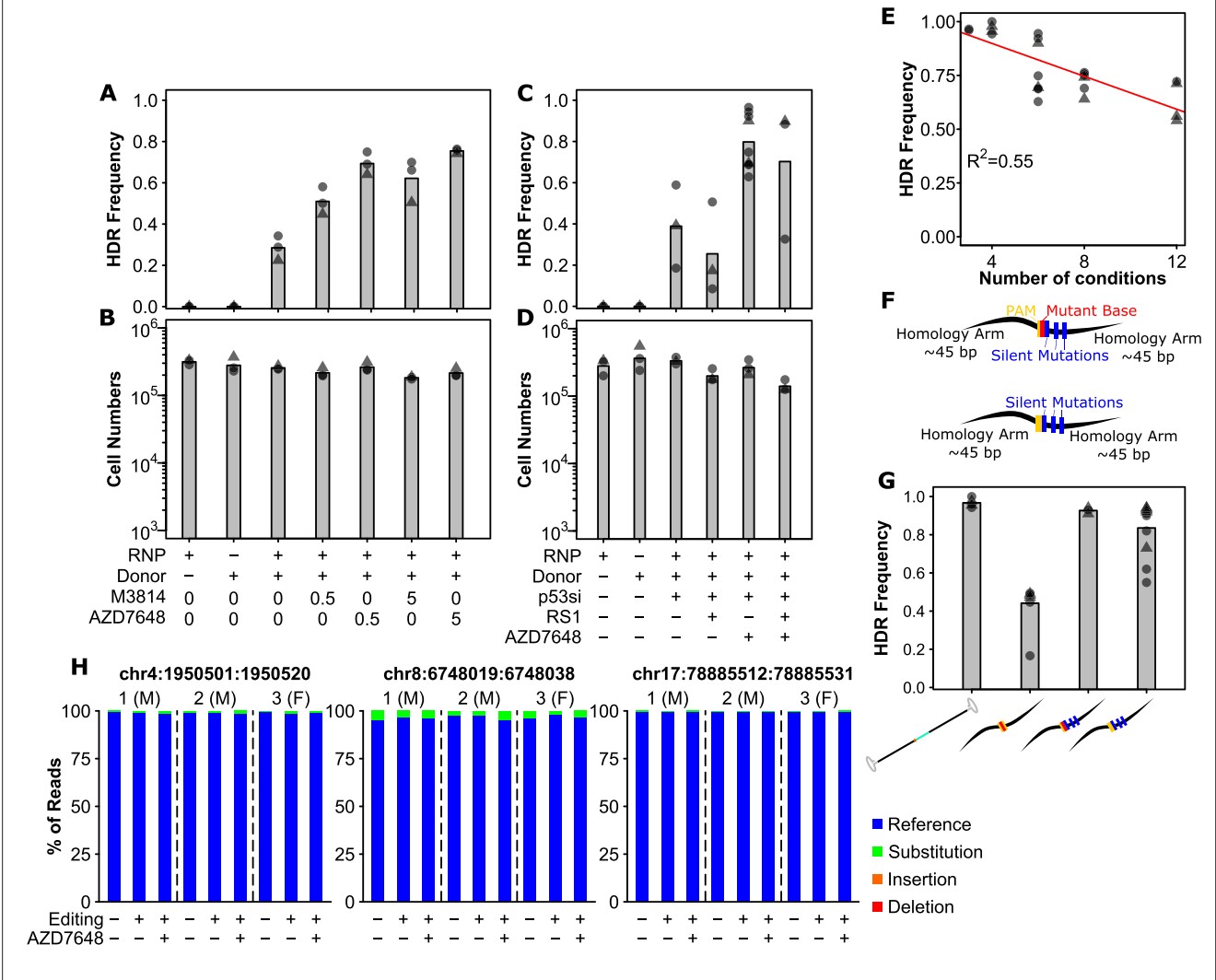

**Figure 2.** Small molecule-mediated inhibition of DNA-PK and optimal donor design substantially improve precise editing efficiency. (**A**) *AZD7648 and M3814 improve homology-directed repair (HDR) efficiency in primary human hematopoietic stem and progenitor cell (HSPC)*. Cells were edited with 30.5 pmol ribonuclear protein (RNP) (or not as indicated) with 400 multiplicity of infection (MOI) of AAV donor and small molecules added as indicated (in µM). Bars show mean values and points show measurements for individual cords. Male cords are shown as triangles and females as circles. (**B**) *Viable cell numbers with AZD7648 and M3814 addition*. Hemocytometer counts at the time of harvest are shown for each sample from (**A**). (**C**) *HDR efficiency with combinations of AZD7648, p53 siRNA, and RS-1*. Cells were edited with 30.5 pmol RNP (or not as indicated) with 400 MOI of AAV donor in the presence of the indicated additives. AZD7648 was used at 5 µM, p53 siRNA at 20 fmol, and RS-1 at 15 µM. (**D**) *Viable cell numbers with additive combinations*. Hemocytometer counts at the time of harvest are shown for each sample from (**C**). (**E**) *Technical factors associated with high sample number is associated with decreased HDR efficiency*. HDR efficiency is shown for all 30.5 pmol RNP, 400 MOI AAV, 20 fmol p53 siRNA, and 5 µM AZD7648 samples by the number of conditions processed in a given experiment. A linear fit is indicated as a red line. The $R^2$ is indicated, and overall p-value was <<0.001. (**F**) *Alternative designs for ssODN donors with key features indicated*. Annotated sequences are shown in Supplementary Information. (**G**) *Silent mutations allow ssODN donors to achieve similar efficiencies to AAV*. All edits were performed with 0.5 µM AZD7648, 20 fmol p53 siRNA, 50 pmol ssODN, or 400 MOI AAV as indicated. Donor types are shown as their logos from (1B, 2F). (**H**) *No observable off-target mutations at predicted target sites even with the addition of AZD7648*. The overall percent of reads containing exclusively reference allele, or any substitutions, deletions, or insertions that overlap with the predicted off-target cut sites is shown for three individual cords across the top 3 cut sites. Cells from each individual cord were split into an unedited control, and cells edited with the silent mutation containing ssODN for the SRSF2 locus under either standard conditions (i.e. no p53siRNA or AZD7648) or with our optimal editing protocol (i.e. with p53siRNA and 0.5 µM AZD7648). False-discovery rate (FDR) corrected paired t-test significance values are shown in *Supplementary file 1*. See also *Figure 2—figure supplements 1–3*.

The online version of this article includes the following figure supplement(s) for figure 2:

**Figure supplement 1.** The addition of AZD7648 also improves editing efficiency at the SF3B1 locus.

**Figure supplement 2.** Example Sanger sequencing traces for the top three predicted off-target sites of the SRSF2 gRNA.

**Figure supplement 3.** Full-length nanopore sequencing traces for the top three predicted off-target sites of the SRSF2 gRNA.

locus (*Figure 1—figure supplement 1C and D*). A closer examination of experimental factors that may affect efficiency revealed this to be related to the number of conditions tested in each experiment (*Figure 2E*). This suggests that there are likely technical factors associated with cell handling and nucleofection timing that play an important role in overall efficiency. Overall, these data suggest that limiting the number of co-processed samples, and the addition of 0.5 μM AZD7648 for the 48 hr following editing could substantially improve editing to near perfect efficiencies.

With our handling and additive conditions, we next tested whether ssODN could be brought to similar efficiencies. Confirming our observed negative correlation between condition number and efficiency, in these tests performed with no more than four simultaneous conditions at nucleofection, we observed a mean efficiency of 97% with AAV donors (*Figure 2F and G*). With the short ssODN, we observed a mean editing efficiency of 44% under these conditions, increased from the 13% but still substantially lower than AAV (*Figure 2F and G*), and consistent with reports for 53BP1 inhibition (*De Ravin et al., 2021*). Further thought into the differences between the two suggested that perhaps it was the further disruption of the spacer sequence provided by the synthetic intron in the AAV donor which provided the difference (*Figure 2F*). As such we redesigned the ssODN donor to incorporate silent mutations throughout the PAM proximal bases of the spacer (*Figure 2F*). We also designed an equivalent donor which did not mutate the PAM but only contained the spacer mutations (*Figure 2F*). Interestingly, the incorporation of these additional spacer-breaking mutations drastically improved editing efficiency up to 94%, nearly that of the AAV (*Figure 2G*, *Figure 1—figure supplement 2C*). Surprisingly, even the design incorporating only the silent spacer mutations, but no PAM mutation achieved a mean efficiency of 84%, further highlighting the importance of spacer mutations in donor design. Importantly, this latter design also contained only silent mutations, selected for codons with equivalent frequency to wild-type, and thus should have no effect on the biology of cells carrying them. Finally, as our protocol temporarily inhibits NHEJ and DNA damage surveillance, we tested whether cells edited under optimal conditions showed an increased off-target editing or not. Analysis of the top 3 predicted off-target sites in 3 sets of samples revealed no detectable edits at these sites either with or without AZD7648 (*Figure 2H*, *Figure 2—figure supplements 2 and 3*). Some substitutions were present both in and around the predicted off-target regions, however, these were equivalent for a given individual between both edited conditions (+/-AZD7568) and unmanipulated controls (paired t-test FDRs >0.3, individual values in *Supplementary file 1*) indicating they were not introduced by the editing process. This suggests that off-target editing at predicted sites is not affected by AZD7648 addition, at least with well-designed gRNA. These modifications to donor design allowed even ssODN to reach near-perfect efficiency in primary human HSPCs.

## Editing efficiency is even across the hematopoietic hierarchy and does not disrupt phenotype proportions or self-renewal and differentiation functions

We next wanted to determine whether editing efficiency was equivalent across the hematopoietic hierarchy, as long-term HSC (LT-HSC) are a minority cell type within the overall CD34 +compartment. While phenotypes involving CD49f are highly selective on fresh cells (*Knapp et al., 2017*; *Notta et al., 2011*; *Knapp et al., 2019*), not all of these markers are stable on cultured cells. As such, we used a panel consisting of only culture-stable markers which retained the ability to sort sub-populations from across the hierarchy. Populations sorted included LT-HSCs (CD34 +CD45RA-CD90+CD49c+), intermediate HSCs (IT-HSC; CD34 +CD45RA-CD90+CD49c-), multipotent and erythroid progenitors (MPP/E; CD34 +CD45RA-CD90-CD49c-) and mature progenitors (Adv-P; CD34 +CD45RA+) (*Tomellini et al., 2019*). Of note, the PAM mutation in our mutant donors introduces a P95H mutation which is associated with myelodysplastic syndrome (MDS), acute myeloid leukemia (AML), and clonal hematopoiesis (*Watson et al., 2020*; *Yoshida et al., 2011*; *van der Werf et al., 2021*). While this mutation is of interest in future disease modelling, to isolate the effects of the editing process itself on HSPC phenotype and function, biological assays were performed exclusively with silent donor. We observed no significant differences in editing efficiencies from bulk measurements regardless of sub-population suggesting that the editing was even across the progenitor hierarchy (*Figure 3A*, *Figure 3—figure supplement 1*). We also observed a generally equivalent representation of each population within the CD34 compartment between silent edited and control cells, though a slight but statistically significant decrease was observed in the proportion of CD34 + cells of the advanced progenitor phenotype (FDR

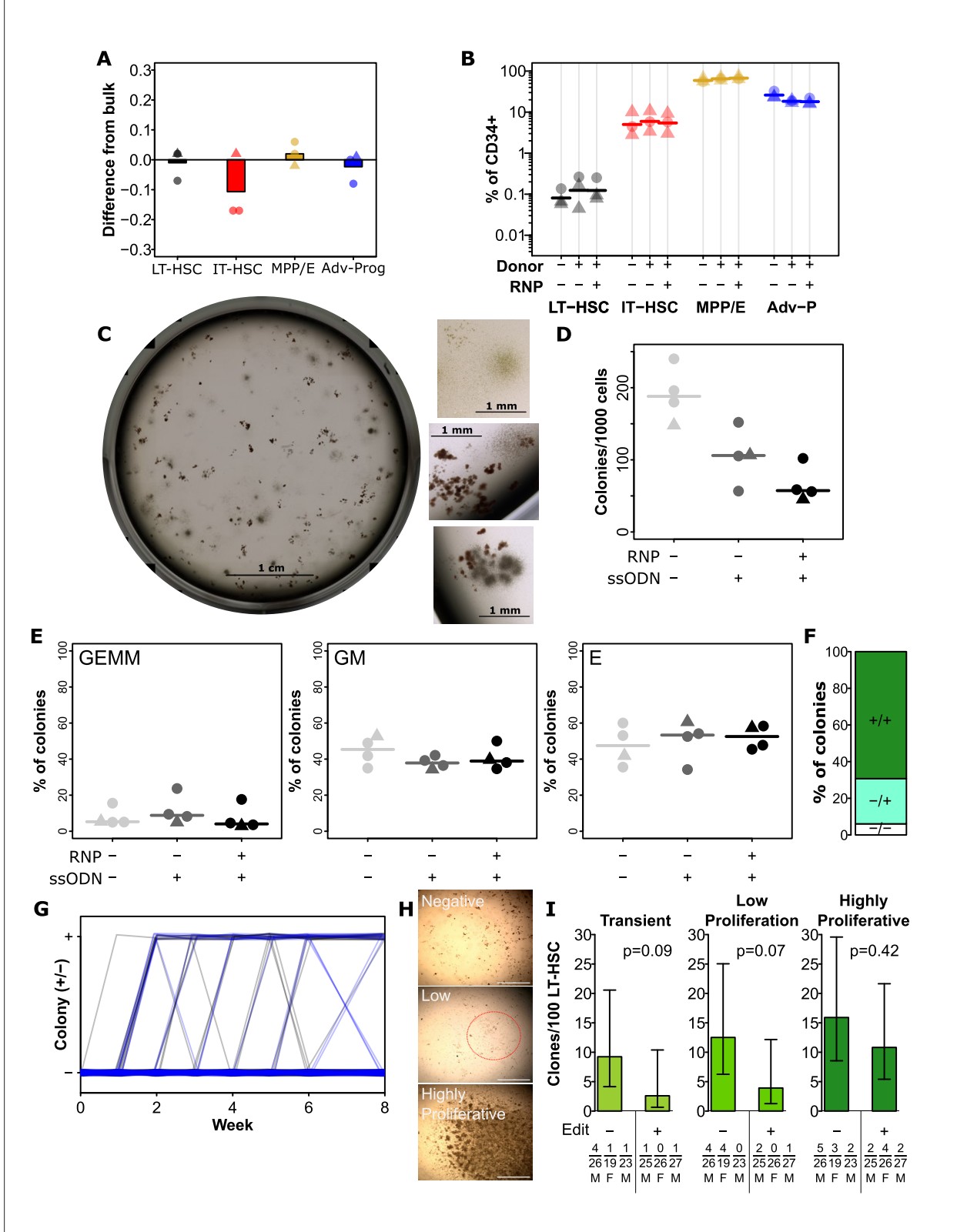

**Figure 3.** Editing has a minimal impact on hematopoietic stem and progenitor cell (HSPC) function and hierarchy. (**A**) *Integration efficiency is equivalent across phenotypically defined progenitor compartments*. All edits were performed with 0.5 μM AZD7648, 20 fmol p53 siRNA, and 50 pmol of silent mutation ssODN. Values show the difference in precise edit efficiency for each phenotypic subset compared to bulk assessment within that cord. Bars show mean values and points show measurements for individual cords. Male cords are shown as triangles and females as circles. All populations show

*Figure 3 continued on next page*

*Figure 3 continued*

no significant difference from bulk. (**B**) *Progenitor phenotypes are minimally altered across the hierarchy*. The % of CD34 + for each sub-population is shown. Mean values are indicated as lines. A slight but significant decrease was present for late progenitors (CD34 +CD45RA+) associated with donor addition (but not different with editing). (**C**) *An example image of a well of colonies, and example colonies*. (**D**) *Total colonies are decreased by the addition of donors, and further by editing*. Total CFC per 1000 CD34 + cells is shown for each cord. Lines indicate mean values. (**E**) *No changes were observed in the frequency of colonies of each type*. As before points are individual cords and lines show mean values. (**F**) *Colonies showed a preponderance of homozygous editing*. Mean homozygous, heterozygous edited, and unedited cells are shown from 36 analyzed colonies across three independent cords. False-discovery rate (FDR) corrected paired t-test significance values are shown in **Supplementary file 1**. (**G**) *No change in the dynamics of colony emergence from single-LT-HSCs in long-term culture initiating cell (LTC-IC)*. The presence or absence of an obvious colony in each well (initially sorted with a single long-term HSC (LT-HSC)) was scored weekly over the first 6 weeks of the LTC-IC assay, and again at week 8. Clonal outputs are shown as lines with unedited in black and edited in blue. (**H**) *Example colonies at 8* weeks. At 8 weeks, clones were scored as negative (no colony at any point), transient (previous colony without a colony at endpoint), low proliferation (>50 cells, but below confluence), and highly proliferative (confluent). Example images of negative, low proliferation, and highly proliferative clones are shown. Scalebars (white) show 1 mm. The low proliferation colony is circled in red. (**I**) *Highly proliferative clones are not lost from the LT-HSC population in the editing process*. The frequency of clones of the indicated types is shown per 100 phenotypic LT-HSC either without editing or following optimal editing. Error bars represent 95% confidence intervals. Frequencies, p-values, and error bars were calculated using Extreme Limiting Dilution Analysis, based on colony numbers measured from three independent experiments (each with a different cord donor). Numbers for each clone per donor, the total number of clones analyzed for that donor, and donor sex are indicated below the relevant bar. See also **Figure 3—figure supplement 1**.

The online version of this article includes the following figure supplement(s) for figure 3:

**Figure supplement 1.** Example gating hierarchy.

= 0.007; **Figure 3B**). To test whether these cells retain proliferative and neutrophil/monocyte/erythroid differentiation potential, we next performed colony-forming cell assays on edited and control cells (**Figure 3C**). Consistent with our earlier observations that donor alone induced some toxicity, we observed a slight drop in colony number even without editing (**Figure 3D**). Colony number was further decreased with editing (**Figure 3D**). This is consistent with the slight drop in the advanced progenitor phenotype which contains primarily neutrophil/monocyte producing progenitors. Importantly, however, no changes to colony type were observed (**Figure 3E**), suggesting that while there is a quantitative drop in progenitor function, this was even across progenitor types and the capability of remaining cells was not affected. Analysis of the editing on colonies revealed editing across most colonies with a predominance of homozygous edits (**Figure 3F**), as expected by our high editing efficiencies.

In order to assess whether the editing process affected the maintenance of HSC with high self-renewal capacity, we used a clonal version of the long-term culture initiating cell (LTC-IC) assay. We have previously shown that clones with high proliferative potential at 8 weeks in this assay correlate with those HSC with the highest regenerative potential in vivo (**Knapp et al., 2018**). Overall, we observed a mixture of negative, transient, low proliferation, and highly proliferative clones across the tested cords and conditions with no difference in the kinetics of clone emergence between edited cells and non-edited controls (**Figure 3G and H**). Critically, there was no significant difference in the frequency of highly proliferative clones between edited cells and non-edited controls (**Figure 3I**), suggesting that the editing process does not substantially affect the most regenerative subset of HSC. While there was not a significant difference, there was a trend towards lower frequencies of low proliferation and transient clones in the edited condition (**Figure 3I**). This is consistent with the observed decrease in CFC frequency (**Figure 3D**), suggesting that later progenitors may be more affected by editing than the most primitive subsets.

## Mutation zygosity can be tuned based on the ratio of mutant to silent donor

While high-level homozygous editing such as those achieved by our current protocol is desired for therapeutic purposes, for disease modelling applications, heterozygosity is often required for accurate modelling. To enable these applications, we tested whether editing efficiency and zygosity could be tuned by providing a mixture of wild-type and mutant donors. To test this, we used a mix of our AAV with different ratios of silent ssODN (**Figure 4A**). At 48 hr post editing a bulk sample was harvested to determine overall editing efficiency. This showed a highly significant monotonic decrease in mutant allele frequency with increasing silent donor proportion (p<<0.001, **Figure 4B**). It should be noted that overall editing efficiencies in these tests were lower than normal due to the requirement

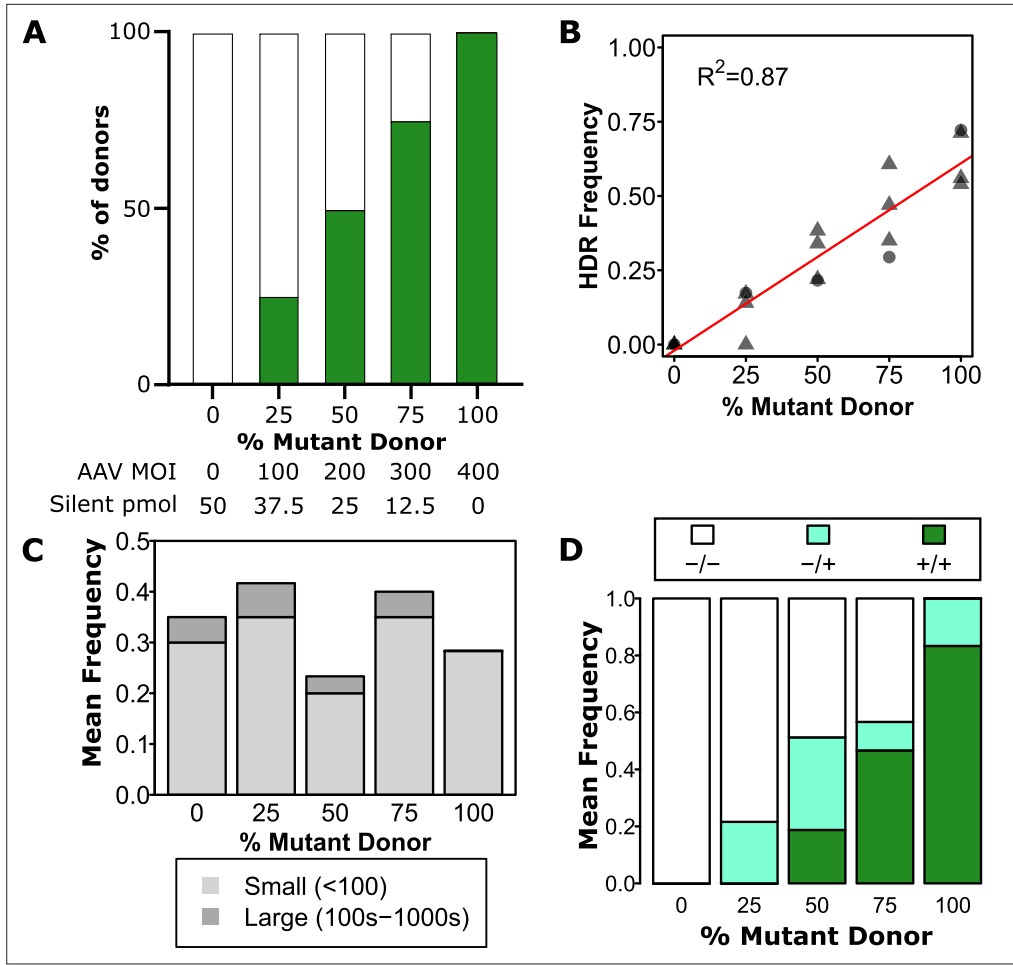

**Figure 4.** Zygosity can be tuned using a mixture of mutant and silent donors. (**A**) *Experimental design.* Green bars represent the proportion of mutant donor. Specific amounts of mutant and silent donor are shown underneath for each condition. All edits were performed with 0.5 μM AZD7648, 20 fmol p53 siRNA, and indicated amounts of each donor. (**B**) *Overall mutant integration efficiency varies linearly with the proportion of mutant donor.* Individual cords are shown as points. Male cords are shown as triangles and females as circles. A linear fit is indicated as a red line. The $R^2$ is indicated, and overall p-value was <<0.001. (**C**) *Mean clonogenic frequencies are consistent across donor proportions.* Data are a mean of two independent cords with a total of 30 cells for each cord at each dose analysed except the 0% condition which only had 20 cells per cord. (**D**) *Zygosity can be adjusted by inclusion of silent donor.* Mean frequencies of homozygous mutant, heterozygous, and homozygous silent donors are shown within all clones with any observed editing. A total of 52 clones with some degree of editing across two independent cords were analyzed.

of simultaneously editing many conditions. To confirm whether the strategy was indeed tuning the zygosity and not simply the overall efficiency, we sorted single CD34 + cells in 96-well plates and allowed these to grow into a colony prior to harvest and genotyping (*Figure 4C*). No significant differences in clonogenic efficiency were observed across conditions though there were no large clones (i.e. in excess of 100 cells) in the 100% mutant donor condition (*Figure 4C*). These results suggest that this analysis was not confounded by differential clonogenicity of specific mutational statuses (homozygous WT/silent, heterozygous, homozygous mutant). As expected, we observed a preponderance of homozygous edits with all mutant donors, with a progressive increase in heterozygous or homozygous silent as the proportion of silent donor was increased (*Figure 4D*). Desired zygosity ratios could thus be predicted based on simple sampling calculations.

## Discussion

The current study demonstrates a near-perfect efficiency of precise editing in primary human HSPC and identifies the key design and technical considerations needed to achieve it. Starting from existing optimized protocols for AAV-based editing (*Bak et al., 2018*), we tested a variety of dosages, HDR donors, donor designs, and small molecule additives for their ability to improve HDR efficiency while retaining cell viability. Through these tests, we confirmed reports that DNA-PK inhibition was an effective method to improve HDR efficiency (*De Ravin et al., 2021*; *Shy et al., 2023*), and identified AZD7648 to be optimal for this task, consistent with demonstrations of its improved potency and specificity (*Fok et al., 2019*). Combining AZD7648 with optimal HDR donors enabled both AAV and ssODN donors to achieve >90% editing efficiency in HSPC. This is particularly interesting as 53BP1 inhibition was reported to be less effective for ssODN donors (*De Ravin et al., 2021*). The difference is likely attributable to the design of the ssODN donors themselves, as we also found that additional silent mutations in the spacer sequence were necessary to achieve maximal efficiencies. Given that most AAV donors insert a tag of some description (*Bak et al., 2018*; *Charlesworth et al., 2018*; *Tran et al., 2022*; *De Ravin et al., 2021*), such a spacer disruption has been built-in to these systems, but not in the competing ssODN. This simple design change facilitates the use of these much cheaper, faster to iterate, and more stable donors. One surprising factor that had a substantial effect on efficiency was the number of conditions simultaneously processed at the time of nucleofection. This likely corresponds to the handling time, and may be analogous to the delivery window for AAV immediately post-nucleofection previously reported (*Charlesworth et al., 2018*).

Beyond simple viability and efficiency, CD34 + cells (HSPCs) are a heterogeneous population made up of a continuous hierarchy of progenitors and stem cells defined by their functional capabilities. To ensure that the editing process was functional throughout the hierarchy, and not specifically toxic to specific levels, we demonstrated that editing did not change the frequency of populations across this hierarchy and that editing efficiency was equivalent throughout, including in the HSC phenotype. These tests made use of phenotypic markers previously shown to be stable and functionally predictive in cultured HSPC (*Tomellini et al., 2019*). Beyond this, using clonogenic assays for progenitor differentiation, we demonstrated that editing did not alter the distribution of myeloid outputs. These findings together with previous reports demonstrating that editing rates were consistent in pre-transplant measured and post-transplantation both with and without DNA-PK inhibition (*Bak et al., 2018*; *Charlesworth et al., 2018*; *Lehnertz et al., 2021*; *De Ravin et al., 2021*) suggest that the reported editing protocol also edited cells capable of engrafting a mouse for >6 months (functional long-term HSC). This is consistent with the maintenance of highly proliferative LTC-IC we observed. That said, long-term transplants were not performed here. It thus remains a formal possibility that functionally defined LT-HSC may either be edited at lower frequency or functionally compromised, though our clonal LTC-IC data would suggest that this is not the case. We did, however, observe a modest decrease in CFC potential and a non-significant trend towards the same in low proliferation LTC-IC, suggesting that late progenitors may be more sensitive to the editing process than the most primitive. This would be consistent with the greater sensitivity of late progenitors to even growth factor stimulation which detrimentally affects their engraftment ability (*Miller et al., 2016*). It is important to note, however, that our and others' data suggest that remaining CFCs appear to retain normal function (*Lehnertz et al., 2021*; *De Ravin et al., 2021*). Interestingly, much of the toxicity both in direct cell survival and CFC numbers could be attributed to donor delivery, regardless of editing. This suggests that toxicity is likely mediated by innate immune responses to DNA (*Piras and Kajaste-Rudnitski, 2021*). Interference with these pathways may thus be a fruitful avenue to mitigate toxicity and should be explored in future studies.

Of course, several other factors remain important considerations for specific use cases. For disease modelling, zygosity can be an important factor, and thus having 100% mutant may not be ideal. To address this issue, we demonstrated that by delivering a combination of silent and mutant donors (of the same strand to prevent donor annealing) one can re-introduce heterozygosity based on the ratio of donors delivered, allowing editing levels and zygosity to be tuned. Another consideration that would be important for any clinical use of edited cells is off-target editing and genomic rearrangements (*Hunt et al., 2023*). We did not observe any off-target edits at predicted sites, regardless of whether edits were done in the presence of AZD7648/p53 siRNA or not, suggesting that the protocol presented here did not substantially increase off-target editing frequencies. This was consistent with

findings from other studies using 53BP1 inhibition which similarly targets NHEJ, where they observed no increases in off target editing or large-scale genomic rearrangements were observed when donor was present (*De Ravin et al., 2021*).

During the time that this article was undergoing its publication process, another manuscript came out showing similar effects for the addition of AZD7648 as being able to increase the precise editing efficiency of human HSPCs (as well as T and B cells, and induced pluripotent stem cells) (*Selvaraj et al., 2024*). In their manuscript they also show an increase in efficiency to similar levels using AAV6 donors, and similar but slightly lower inefficiencies that we achieved in ssODN donors (*Selvaraj et al., 2024*). The differences are likely attributable to our use of shorter ssODN donors and our choice of silent spacer disrupting mutations in them. Both of these factors have been previously shown to impact the integration efficiency (*Okamoto et al., 2019*). We also observed similar viability effects of AAV donor, and lack of negative effects from 0.5 µM AZD7648 (*Selvaraj et al., 2024*), though we extend this with the effects of the ssODN donors and ssODN donor length on viability. Again, we similarly show that CFC distributions are not affected by the editing process (*Selvaraj et al., 2024*), though we again extend this with information on the effect of ssODN donors on total CFC frequency. Beyond what was done in the other manuscript, we also show that the editing process and addition of AZD7648 edits evenly across the HSPC hierarchy, and that it does not substantially affect the distribution of progenitor phenotypes and critically does not affect the LTC-ICs with the highest self-renewal potential. Also unique in our manuscript, we demonstrate that for use in mutation modeling/leukemia where zygosity can be an important consideration, it's possible to tune the zygosity of output cells by adjusting the ratios of mutant to silent donors. Finally, another article was also published while we were in our review process in which they used AZD7648 together with Polθ inhibitors in several non-hematopoietic cell lines to improve editing efficiency (*Wimberger et al., 2023*). Overall, these and our manuscripts cross-validate the use of AZD7648 for improving precise genome editing in HSPC and beyond (and across more loci, as ours differ completely from theirs *Selvaraj et al., 2024*; *Wimberger et al., 2023*).

Overall, this study identified a number of critical factors that determine the efficiency of precise editing in primary human HSPCs, and how the combination of these can allow editing at near-perfect efficiency. Moreover, this efficiency could be tuned by the inclusion of competing silent donor. We anticipate that the protocol presented here will enable precise isogenic disease modelling for a variety of monogenic blood diseases directly in primary human HSPCs. The principles and protocols will also permit further improvement in existing approaches for therapeutic editing that are even now in trial.

# Materials and methods

### Key resources table

| Reagent type (species) or resource | Designation | Source or reference | Identifiers | Additional information |
|---|---|---|---|---|
| Antibody | AF647 (mouse monoclonal) Anti-Human CD34 (clone 581) | Cedarlane | 343508 | 1:200 |
| Antibody | V450 (mouse monoclonal) Anti-Human CD45RA (clone HI100) | BD Biosciences | 560362 | 1:100 |
| Antibody | PE-CF594 (mouse monoclonal) Anti-Human CD90 (clone 5E10) | BD Biosciences | 562385 | 1:200 |
| Antibody | FITC anti-human, CD49c (Clone REA360) | Miltenyi | 130-105-364 | 1:50 |
| Genetic reagent (AAV2/6) | Custom AAV6 – pssAAV_SRSF2P95H | Canadian Neurophotonics Platform – Viral Vector Core | Custom AAV | See annotated sequences in *Supplementary file 2* |
| Genetic reagent (AAV2/6) | Custom AAV6 – pssAAV_SF3B1 K700E | Canadian Neurophotonics Platform – Viral Vector Core | Custom AAV | See annotated sequences in *Supplementary file 2* |
| Biological sample (*Homo sapiens*) | Human Cord Blood for CD34 + cells harvest | Héma Québec via St Justine hospital | NA | |
| Cell line (*Mus musculus*) | M210B4 expressing human IL-3 and G-CSF | Gift from Connie J Eaves | NA | *special request |

*Continued on next page*

*Continued*

| Reagent type (species) or resource | Designation | Source or reference | Identifiers | Additional information |
|---|---|---|---|---|
| Cell line (*Mus musculus*) | sl/sl mouse fibroblasts expressin human SCF and IL-3 | Gift from Connie J Eaves | NA | *special request |
| Cell line (*Mus musculus*) | sl/sl mouse fibroblasts expressin human FLT3L | Gift from Connie J Eaves | NA | *special request |
| Chemical compound, drug | UM 171 | ExcellThera | NA | *special request |
| Chemical compound, drug | RS-1 | Cedarlane | 21037–5 | |
| Chemical compound, drug | Nedisertib (M3814) | Cedarlane | A17055 | |
| Chemical compound, drug | AZD 7648 | Cedarlane (Cayman) | 28598–1 | |
| Chemical compound, drug | DIMETHYL SULFOXIDE (DMSO), Sterile | BioShop | DMS666.100 | |
| Chemical compound, drug | Hydrocortisone | BioShop | HYD400.5 | |
| Chemical compound, drug | 1 M buffer | Homemade | Homemade | |
| Commercial assay, kit | PLATINUM SUPERFI II MASTER MIX | Life Technologies | 12368050 | |
| Commercial assay, kit | StemSpan CC100 | STEMCELL Technologies | 2690 | |
| Commercial assay, kit | MyeloCult H5100 | STEMCELL Technologies | 05150 | |
| Commercial assay, kit | Blunt/TA Ligase Master Mix | NEB | M0367S | |
| Commercial assay, kit | NEBNext Quick Ligation Module | NEB | E6056S | |
| Commercial assay, kit | NEBNext Ultra II End Repair/ dA-Tailing Module | NEB | E7546S | |
| Commercial assay, kit | EasySep Human CD34 Positive Selection Kit II | STEMCELL Technologies | 17896 | |
| Commercial assay, kit | MethoCult H4034 Optimum | STEMCELL Technologies | 4034 | |
| Commercial assay, kit | Flongle Sequencing Expansion | Oxford Nanopore | FLO-FLG114 | |
| Commercial assay, kit | Flongle Flow Cell (R10.4.1) | Oxford Nanopore | FLO-FLG114 | |
| Commercial assay, kit | Native Barcoding Kit 24 V14 | Oxford Nanopore | SQK-NBD114.24 | |
| Commercial assay, kit | FBS Canadien | Thermo | 12483020 | |
| Commercial assay, kit | RPMI1640 | LifeTech | 11875119 | |
| Commercial assay, kit | StemSpan SFEM II | STEMCELL Technologies | 9655 | |
| Peptide, recombinant protein | Alt-R S.p. Cas9 Nuclease V3, 500 µg | IDT | 1081058 | |
| Peptide, recombinant protein | T7 Endonuclease I - 250 units | NEB | M0302S | |
| Peptide, recombinant protein | Proteinase K, Molecular Biology Grade | NEB | P8107S | |
| Peptide, recombinant protein | Alt-R Cas9 Electroporation Enhancer, 2 nmol | IDT | 1075915 | |
| Peptide, recombinant protein | BspEI Enzyme | NEB | R0540S | |
| Peptide, recombinant protein | IL-3, Human (CHO-expressed), 100 ng/ul | Cedarlane (GeneScript) | Z02991-10 | |
| Peptide, recombinant protein | SCF, Human (P. pastoris-expressed), 100 ng/ul | Cedarlane (GeneScript) | Z02692-10 | |

*Continued*

| Reagent type (species) or resource | Designation | Source or reference | Identifiers | Additional information |
|---|---|---|---|---|
| Peptide, recombinant protein | EPO 100 ng/ul (~16 IU/uL) | Cedarlane (GeneScript) | Z02975-10 | |
| Peptide, recombinant protein | Flt-3L 100 ng/ul | Cedarlane (GeneScript) | Z02926-10 | |
| Peptide, recombinant protein | GM-CSF, Human (CHO-expressed), 100 ng/uL | Cedarlane (GeneScript) | Z02983-10 | |
| Peptide, recombinant protein | IL-6, Human (CHO-expressed), 100 ng/uL | Cedarlane (GeneScript) | Z03134-50 | |
| Peptide, recombinant protein | G-CSF, Human (CHO-expressed), 100 ng/uL | Cedarlane (GeneScript) | Z02980-10 | |
| Peptide, recombinant protein | CellAdhere Type I Collagen, Bovine, Solution | STEMCELL Technologies | 7001 | |
| Recombinant DNA reagent | p53 siRNA id s605 | Thermo | 4390824 | |
| Sequence-based reagent | SRSF2_gRNA1 | IDT | /AltR1/rCrGrGrCrUrGrUrG rGrUrGrUrGrArGrUrCrCr GrGrGrGrUrUrUrArGrAr GrCrUrArUrGrCrU/AltR2/ | crRNA SRSF2 |
| Sequence-based reagent | pri0077-F | IDT | AGCGATATAAACGGGCGCAG | Outer PCR SRSF2 |
| Sequence-based reagent | pri0077-R | IDT | TCGCGACCTGGATTTGGATT | Outer PCR SRSF2 |
| Sequence-based reagent | pri0002-H3 | IDT | CTATGGATGCCATGGACGGG | Inner PCR SRSF2 |
| Sequence-based reagent | pri0002-H4 | IDT | CAAGCACAGCGGGGTTAATTC | Inner PCR SRSF2 |
| Sequence-based reagent | pri0261-F | IDT | TCATTGGCAAACAGCAAGCC | SRSF2 gRNA1 off-target 1 |
| Sequence-based reagent | pri0261-R | IDT | AGAAGTATGTGCCTACGCGG | SRSF2 gRNA1 off-target 1 |
| Sequence-based reagent | pri0262-F | IDT | GAGAGTCACCGACCATGACG | SRSF2 gRNA1 off-target 2 |
| Sequence-based reagent | pri0262-R | IDT | TGTAAAACGTGCTGGAGGCT | SRSF2 gRNA1 off-target 2 |
| Sequence-based reagent | pri0263-F | IDT | CAGAAAGCACAAGCAACGCT | SRSF2 gRNA1 off-target 3 |
| Sequence-based reagent | pri0263-R | IDT | TCTCTTCCGGACACAAGTGC | SRSF2 gRNA1 off-target 3 |
| Sequence-based reagent | pri0285 | IDT | CTCCTTCTTCACGTCTTCCT | SRSF2 off-target 2 sequencing |
| Sequence-based reagent | pri0286 | IDT | CACCACATCTGGGATCCTCA | SRSF2 off-target 3 sequencing |
| Sequence-based reagent | SF3B1 Cas9 gRNA K700 | IDT | /AltR1/rUrGrGrArUrGrArGrCrArGr CrArGrArArArGrUrGrUrGrUrUrAr GrArGrCrUrArUrGrCrU/AltR2/ | crRNA SF3B1 |
| Sequence-based reagent | Alt-R CRISPR-Cas9 tracrRNA | IDT | 1072533 | tracrRNA |
| Sequence-based reagent | pri0078-F | IDT | GCTGCTGGTCTGGCTACTAT | Outer PCR SF3B1 |
| Sequence-based reagent | pri0078-R | IDT | ATACTCATTGCTGATTACGTGATTT | Outer PCR SF3B1 |
| Sequence-based reagent | pri0002-H1 | IDT | TGGGCTACTGATTTGGGGAG | Inner PCR SF3B1 |

*Continued on next page*

*Continued*

| Reagent type (species) or resource | Designation | Source or reference | Identifiers | Additional information |
|---|---|---|---|---|
| Sequence-based reagent | pri0002-H2 | IDT | CTGTGTTGGCGGATACCCTT | Inner PCR SF3B1 |
| Sequence-based reagent | SRSF2 Silent ssODN | IDT | t*g*gacggccgcgagctgcgggtgcaaatg gcgcgctacggccgcccTccAgaTtcacacca cagccgccggggaccgccacccccgcag*g*t | |
| Sequence-based reagent | SRSF2 P95H ssODN | IDT | t*g*gacggccgcgagctgcgggtgcaaatggcg cgctacggccgccATccggactcacaccacag ccgccggggaccgccacccccgcag*g*t | |
| Sequence-based reagent | SRSF2 long ssODN donor | IDT | Ttcacgacaagcgcgacgctgaggacgctatgga Tgccatggacggggccgtgctggacggccgcga gctgcgggtgcaaatggcgcgctacgg ccgccATccggactcacaccacagccgccggg gaccgccaccccgcaggtacgggggcggtggcta cggacgccggagccgcaggtaaacgg ggctgaggggaccg | ordered as ALT-R HDR Donor Oligo |
| Sequence-based reagent | SRSF2 P95H ssODN with additional silent mutations | IDT | t*g*gacggccgcgagctgcgggtgcaaatggcgcgctac ggccgccATccAgaTtcacaccacagccgccggg gaccgccacccccgcag*g*t | |
| Software and algorithms | GelAnalyzer 19.1 | Istvan Lazar Jr. and Istvan Lazar Sr. | https://www.gelanalyzer.com | |
| Software and algorithms | Synthego Performance Analysis V3 | ICE Analysis | https://www.synthego.com | |
| Software and algorithms | FlowJo Software 10.8.1 | BD Life Sciences | https://www.flowjo.com | |
| Software and algorithms | R (version 4.1.2) | R Core Team | https://www.r-project.org/ | |
| Software and algorithms | MinKNOW (version 23.11.3) | Oxford Nanopore | https://community.nanoporetech.com/downloads | |
| Software and algorithms | minimap2 (version 2.26) | Dana-Farber Cancer Institute | https://github.com/lh3/minimap2 | |
| Software and algorithms | samtools (version 1.10) | Genome Research Ltd. | http://www.htslib.org/ | |
| Software and algorithms | bcftools (version 1.10.2) | Genome Research Ltd. | https://samtools.github.io/bcftools/ | |
| Software and algorithms | Biopython (version 1.83) | Biopython | https://biopython.org/ | |
| Software and algorithms | Python (version 3.9.7) | Python | https://www.python.org/ | |

## Human cord blood processing and cryopreservation

Anonymized consented human umbilical cord blood was obtained from Hôpital St-Justine and Hema Quebec, Montréal, QC, Canada. Ethics approval for the use of these cells was granted from the Comité d'éthique de la recherche clinique (CERC) of the Université de Montréal. CD34 + cells were isolated using EasySep Human CD34 Positive Selection Kit II (STEMCELL Technologies, Vancouver, BC, Canada) as per manufacturer instructions, and then used either directly or cryopreserved in fetal bovine serum (Gibco) and 10% dimethyl sulfoxide (BioShop). Cells were frozen slowly at –80 °C in a CoolCell (Corning) and transferred the following day to liquid nitrogen until use. On thaw, cells were rapidly warmed to 37 °C in a water bath, diluted 10 x in RPMI + 10% FBS (Gibco), and spun to remove media and residual DMSO. Viable cells were counted using a hemocytometer with Trypan Blue (Gibco) and placed into culture.

## CD34+ cell culture

CD34 + cells were cultivated in StemSpan SFEM II media supplemented with 1 X StemSpan CC100 and 35 nM UM171. Cell density was maintained below 250,000 cells/mL of total media to prevent auto-inhibition of the primitive cells (*Knapp et al., 2017*; *Csaszar et al., 2012*). Cells were maintained

in a humidified incubator at 37 °C with periodic viable cell counts (by hemocytometer) to ensure that the density remained within the acceptable range.

## RNP assembly

The tracrRNA and crRNA (IDT) were mixed at equimolar ratio to a final concentration of 100 μM, annealed for 5 min at 95 °C and cooled to 25 °C at 0.1 °C/s. Annealed gRNA was then added to Cas9 enzyme (IDT) and incubated at room temperature for 15 min with a ratio of Cas enzyme to gRNA of 1:2.5.

## CD34+ cell editing

After 48 hr of pre-stimulation, viable cells were counted by hemocytometer. Prior to nucleofection, cells were washed once with PBS, spun down for 5 min at 300 g, and re-suspended in buffer 1M (*Chicaybam et al., 2016*) (5 mM KCl; 15 mM MgCl2; 120 mM Na2HPO4/NaH2PO4 pH7.2; 50 mM Manitol) such that each well of the Nucleocuvette strip would contain 20,000–100,000 cells. Assembled RNP, p53 siRNA (20 fmol, Thermo id s605), electroporation enhancer (IDT), and any ssODN donors (IDT) (as specified in each experiment). Overall RNP and other additives were kept at or below 10% of the total 20 μL volume per well. Handling time between wash and nucleofection was kept within a 10 min window. Cells were nucleofected using the Lonza 4D nucleofector device with nucleocuvette strips, Primary P3, and DZ100 program. Following nucleofection, cells were allowed to rest for 5 min and then added to pre-warmed wells of a 24-well plate containing media (as specified in CD34 + cell culture) supplemented with small molecules as indicated for specific experiments (AZD7648, RS-1, Cayman Chemicals; M-3814, Toronto Research Chemicals). Where AAV donor was used, it was added to the well within 15 min of nucleofection (*Charlesworth et al., 2018*). Custom AAV donors were generated by the Canadian Neurophotonics Platform Viral Vector Core Facility (RRID:SCR_016477). Cells were incubated for an additional 48 hr prior to subsequent use. After this 48 hr period, viable cells were counted by hemocytometer with Trypan Blue.

## Genomic DNA lysis and amplification

At time of harvest, cells were centrifuged at 300 g for 5 min and washed once with PBS, and pelleted again followed by re-suspension in a gDNA lysis buffer (50 mM Tris, 1 mM EDTA, 0.5% Tween-20 and 16 U/mL Proteinase K). Samples were incubated 1 hr at 37 °C and 10 min at 95 °C. Lysates were amplified by PCR using the Platinum Taq SuperFi II Master Mix with 0.5 μM of each primer, and lysate comprising no more than 5% of the total volume of the reaction. For T7E1 assays, short, and long ssODN without silent mutations, a single amplification was performed with primers pri0002-H3+pri0002 H4 (Key Resource Table). For short ssODN with silent mutations, a single amplification was performed with pri0077-F+pri0077 R (Key Resource Table). For AAV, a nested PCR was performed with where the first PCR was performed (SRSF2=pri0077 F+pri0077 R, SF3B1=pri0078 F+pri0078 R; Key Resource Table) followed by purification using a GeneJet PCR purification kit (Thermo) and 15–30 ng transferred to a second PCR (SRSF2=pri0002 H3+pri0002 H4, SF3B1=pri0002 H1+pri0002 H2; Key Resource Table). For analysis of off-target cutting, the top three sites predicted in Benchling were amplified with primers as indicated in the Key Resource Table and sequenced using either indicated sequencing primers (off-targets 2 & 3) or one of the amplification primers (off-target 1). For all PCRs, the program was 98 °C 30 s; 35x (98 °C 10 s, 60 °C, 10 s, 72 °C 30 s); 72 °C 5 min.

## Measurement of RNP cutting efficiency after 48 hr post-electroporation using the T7E1 assay

PCR products were purified using the GeneJet PCR purification kit, and DNA quantified by nanodrop. 100 ng of DNA was added to 1 x NEBuffer 2, and annealed as follows: 95 °C 5 min; 95–85°C–2°C/s; 85–25°C–0.1°C/s; hold at 4 °C. The sample was then split in half, with half kept aside as a non-digested control, and 2.5 U T7 Endonuclease I added to the other half. All samples are incubated at 37 °C for 15 min and then run on a 2% TAE agarose gel with GelRed (Biotium). Gels were imaged at non-saturating intensities using a GeneGenius Imaging System (Syngene). The ratio of cut to uncut bands was then quantified by densitometry.

## Quantification of HDR donor integration

HDR donor integration was assessed by three different ways depending on the HDR donor. For the PAM mutant only short and long ssODNs, a BspEI site was introduced by the mutation. For these,

100 ng of amplified DNA was added to 1 x NEBuffer 3.1 with 2 U BspEI. These were then incubated at 37 °C for 1 hr and then run on a 2% TAE agarose gel with GelRed and quantified by densitometry. For AAV donors, these were run directly on a gel, and integration quantified based on the size shift from the introduced 143 bp synthetic intron. Finally, for ssODN with silent mutations, PCR products were purified by GeneJet PCR purification, quantified by nanodrop, and 5–15 ng sent for Sanger sequencing at the IRIC Genomics core with the primer pri0003-A1 (Key Resource Table). Integration was then quantified using ICE Analysis (Synthego Performance Analysis, 2019. v3.0. Synthego) with a comparison between each edited sample back to a matched unedited control.

### Analysis of off-target editing by nanopore sequencing

Amplicons for each of the three predicted off-target sites were first pooled per sample at equimolar ratios based on their known length and concentration measured by Nanodrop. This allows higher multiplexing without requiring additional barcoding reagents as each amplicon will map uniquely. Each off-target pool was then end-repaired using the NEBNext Ultra II End Repair/dA-Tailing Module with the addition of DNA Control Sample (Oxford Nanopore) as per manufacturer instructions, and end-repaired/A-tailed products purified using AmpureXP beads at a 1 x bead ratio. Unique native barcodes were then added to each repaired/tailed amplicon pool using the Native Barcoding Kit 24 V14 (Oxford Nanopore) and NEB Blunt/TA Ligase Master Mix as per manufacturer instructions, all barcoded amplicons pooled into a single tube, and the library purified using AmpureXP beads, this time at a 0.7 x bead ratio. Finally, sequencing adapters were added to the pooled library using the NEBNext Quick Ligation Module (NEB) with the native adapter from the Native Barcoding Kit 24 V14, and this again purified with AmpureXP beads using Short Fragment Buffer (Oxford Nanopore) instead of 80% ethanol, all as per manufacturer instructions. Final concentration was determined by Nanodrop and 20 fmol loaded onto a Flongle Flow Cell (R10.4.1) with a miniON sequencing device (MIN-101B) with Flongle adapter (all from Oxford Nanopore). Samples were run with MinKNOW, and base-calling executed on Super-Accurate mode.

Following base-calling, a custom pipeline based on bcftools was used to call variants (available at: https://github.com/djhfknapp/Nanopore_Amplicon_CRISPR_Analysis; copy archived at **Knapp, 2024**). For this, fastq files were first pooled per barcode using Biopython. These were then aligned to the known amplicon sequences (from GRCh38) using minimap2 with the map-ont option, converted to a BAM file, sorted, and indexed using samtools view, sort, and index, respectively. Variants were then called using bcftools mpileup with options '-Ov -Q 16 a FORMAT/DP,FORMAT/AD -d 10000 --min-MQ 10'. Next a per site summary was generated from the VCF file counting all high-quality (those meeting the bcftools criteria, i.e. base quality ≥16, mapping quality ≥10) reference reads, substitutions, deletions, and insertions, and output as a TSV file. This output was used directly for per-sample plotting. For summary statistics, the predicted target sites were identified for each amplicon, and the percentage of reads supporting specific substitutions, deletions (starting in or extending into the target site), or insertions were calculated per site. Only variants supported by at least 3 reads and making up at least 0.1% of total reads at a site were counted for these summary statistics. Assuming that any given read has only a single variant which impacts the target region, fully reference percentages were then calculated as 100 minus the sum of all substitutions, deletions, and insertions. This assumption was generally valid with the target sequences used and if anything would under-represent percent of fully reference matching reads.

### Gel densitometry

Band quantification was performed using GelAnalyzer 19.1 (https://www.gelanalyzer.com). Briefly, lanes were detected using the detect bands on every lane function. Peaks were then detected automatically using the 'Detect Bands on Every Lane' function. Background was subtracted using the Rolling Ball option within autodetect background on all lanes with a peak width tolerance of 15%. The contribution of each band was then calculated as the density of that peak divided by the total density of all peaks.

### Flow cytometry and sorting

48 hr post-edit, cells were collected, spun down, and stained in PBS + 2% FBS using a panel of four antibodies: CD34 (1:200, AF647 Mouse Anti-Human CD34 (clone 581)), CD45RA (1:100, V450

Mouse Anti-Human CD45RA (clone HI100)), CD90 (1:200, PE-CF594 Mouse Anti-Human CD90 (clone 5E10)) and CD49c (1:50, FITC anti-human, CD49c (Clone REA360)) for 1 hr on ice in the dark. Precision Count Beads (BioLegend) were added prior to sort to allow quantification of absolute numbers. Cells were analyzed and sorted on a BD FACSAria III sorter. Cells were either sorted for CD34 +for colony-forming cell (CFC) assays, or for four sub-populations as follows: long-term HSCs (LT-HSC; CD34 +CD45RA-CD90+CD49c+), Intermediate HSCs (IT-HSC; CD34 +CD45RA-CD90+CD49c-), multi-potent and erythroid progenitors (MPP/E; CD34 +CD45RA-CD90-CD49c-), and late progenitors (Adv-P; CD34 +CD45RA+). In all cases, this was done on purity mode using dilute samples to minimize cell lose and prevent possible contamination. FACS data was analysed using FlowJo software (version 10.8.1).

## CFC assays

For CFC assays, 500 and 1500 CD34 + cells were added to 1 mL of MethoCult H4034 Optimum medium (StemCell Technologies) and placed into one well of a six-well plate. Plates were incubated for 14–16 days in a humidified incubator at 37 °C, 5% $CO_2$. Wells were then imaged in colour using a Cytation 5 (BioTek), and colonies scored based on colour, morphology, and size as previously performed (**Knapp et al., 2019**). Selected colonies were then picked into PBS, spun down, and subjected to gDNA extraction, PCR, and integration assessment as detailed above.

## Long-term culture initiating cell (LTC-IC) assays

Cells were either edited with silent SRSF2 ssODN donors as per our optimal protocol: 48 hr pre-stimulation in StemSpan+CC100+35 nM UM171, delivery of editing components by nucleofection of 30.5 pmol RNP +50 pmol ssODN +20 fmol p53siRNA, and 48 hr culture in StemSpan +CC100+35 nM UM171+0.5 μM AZD7648, or controls maintained in StemSpan +CC100+35 nM UM171 for the 4 day period. At the end of this period, single LT-HSC (CD34 +CD45RA-CD90+CD49c+) were sorted per well into the inner 60 wells of a flat-bottom 96-well LTC-IC plate. Edited cells were sorted into half of each plate and unedited controls into the other half. LTC-IC assays were performed as described in **Knapp et al., 2018**. Briefly, one day before sorting, each well of the 96-well plate was first coated with 45 μL 2.25% Type I Bovine Collagen Solution (StemCell Technologies) for 1 hr at 37 C. Next, $5 \times 10^4$ cells comprising an equal mixture of irradiated M210B4 fibroblasts expressing human IL-3 and G-CSF, sl/sl mouse fibroblasts expressing human SCF and IL3, and sl/sl mouse fibroblasts expressing human FLT3L were added to each well. Feeder cells were received pre-mixed and irradiated from Dr. Connie J Eaves. Batches were pre-tested at the source lab for cytokine production by qPCR and ELISA, and for Mycoplasma using a VenorGeM kit. The day of the sort, media was changed to 100 μL Myelocult H5100 (StemCell Technologies) supplemented with $10^{-6}$ M hydrocortisone (BioShop). Following the sort, cultures were maintained at 37 °C in a humidified incubator with 5% CO2 with weekly half-media changes for 6 weeks. Each week wells were imaged on a Cytation 5 imager, and scored for the presence of refractile non-adherent cells. At the end of 6 weeks, cultures were supplemented with 50 ng/mL recombinant human SCF +20 ng/mL each of GM-CSF, IL-3, IL-6, G-CSF, and 3 U/mL erythropoietin (all from Genscript). Cultures were allowed to continue for an additional 2 weeks and then imaged and scored again. For this final timepoint, confluent cultures were scored as highly proliferative, those with >50 cells but below confluence were scored as low proliferation, those which had detectable cells at earlier time points but no longer at the final scoring were scored as transient, and those in which no clones were ever detected were scored as negative.

## Single-cell cloning for zygosity analyses

Single CD34 +CD45RA- cells were sorted (see staining and sort details above) 48 h post-editing on single cell mode into independent wells of a 96-well plate and cultured as detailed in CD34 + cell culture for up to 14 days. When a clone reached a large size (100 s-1000s of cells) they were harvested, or at the end of the 14 day period those with visible but small clones (<100 cells) were harvested. In each case, these underwent lysis, PCR, and gel assessment. All integration negative colonies were assessed for silent donor integration by Sanger sequencing/ICE analysis as described above.

## Statistical analyses

All statistical analyses were performed in R (version 4.1.2). For statistical comparisons, paired t-tests were performed in cases where the same cord was used for all conditions and unpaired where necessary. In all cases, an FDR correction for multiple testing was performed using the 'p.adjust' function. For LTC-IC frequencies, Extreme Limiting Dilution Analysis was performed using R function 'elda' from the package 'statmod'.

## Acknowledgements

Funds for this work were provided by the IRIC Philanthropic funds from the Marcelle and Jean Coutu foundation, Fonds Innovation du Groupe Canam, the Cancer Research Society (Operating Grants program #944254), and the Terry Fox Research Institute (Terry Fox New Investigator Award #TFRI 1118). DJHFK has salary support from FRQS in the form of a Chercheurs-boursiers Junior 1 fellowship (#283502). FMCU was supported by a Cole Foundation Doctoral Award, a bourse d'excellence du programme de biologie moléculaire from the Université de Montréal and a PhD scholarship from the Institut de Recherche en Immunologie et en Cancérologie.

The authors would like to thank Thomas Sontag and Michel Duval from the Banque de Recherche de Sang de Cordon at CHU Sainte-Justine and HemaQuebec for assistance with cord blood acquisition, the generous individuals who donated their cords for this project, Annie Gosselin and Angélique Bellemare-Pelletier from the IRIC Flow Cytometry facility for technical assistance with sorts, Raphaelle Lambert from the IRIC Genomics facility for technical assistance with Sanger sequencing, the Canadian Neurophotonics Platform Viral Vector Core Facility (RRID:SCR_016477) for assistance with AAV generation. AAV2_HLF-ZE which was used to generate the SRSF2 AAV donor was a gift from Guy Sauvageau (Addgene plasmid # 175034; http://n2t.net/addgene:175034; RRID:Addgene_175034). The LTC-IC feeders were a generous gift from Dr. Connie J Eaves provided with the assistance of Glenn Edin and Margarita E MacAldaz.

## Additional information

### Competing interests

Bernhard Lehnertz: BL is currently the director of Stem Cell Engineering at ExcellThera. ExcellThera was not involved in the design, execution, or interpretation of the current study. Guy Sauvageau: GS is co-CEO of ExcellThera. ExcellThera was not involved in the design, execution, or interpretation of the current study. The other authors declare that no competing interests exist.

### Funding

| Funder | Grant reference number | Author |
|---|---|---|
| Cancer Research Society | 944254 | David JHF Knapp |
| Terry Fox Research Institute | TFRI 1118 | David JHF Knapp |
| Fonds de Recherche du Québec - Santé | 283502 | David JHF Knapp |

The funders had no role in study design, data collection and interpretation, or the decision to submit the work for publication.

### Author contributions

Fanny-Mei Cloarec-Ung, Conceptualization, Data curation, Investigation, Visualization, Methodology, Writing – original draft, Writing – review and editing; Jamie Beaulieu, Arunan Suthananthan, Investigation, Writing – review and editing; Bernhard Lehnertz, Resources, Methodology, Writing – review and editing; Guy Sauvageau, Resources, Supervision, Writing – review and editing; Hilary M Sheppard, Conceptualization, Supervision, Methodology, Writing – review and editing; David JHF Knapp, Conceptualization, Formal analysis, Supervision, Funding acquisition, Validation, Visualization, Methodology, Writing – original draft, Project administration, Writing – review and editing

## Author ORCIDs
Fanny-Mei Cloarec-Ung ⬡ http://orcid.org/0009-0007-0822-0920
David JHF Knapp ⬡ https://orcid.org/0000-0002-2161-9959

## Ethics

Human subjects: Anonymized consented human umbilical cord blood was obtained from Hôpital St-Justine and Hema Quebec, Montréal, QC, Canada. Ethics approval for the use of these cells was granted from the Comité d'éthique de la recherche clinique (CERC) of the Université de Montréal.

Reviewer #2 (Public review): https://doi.org/10.7554/eLife.91288.3.sa1
Author response https://doi.org/10.7554/eLife.91288.3.sa2

---

## Additional files

### Supplementary files
• Supplementary file 1. Significance testing. Figure and panel are indicated along with each pairwise test. Where relevant the colony type or population for a given test is indicated.

• Supplementary file 2. Annotated DNA sequence used in the experiments for HDR testing and integration.

• MDAR checklist

• Source data 1. Raw gel images. Data file for the raw gels compiled in *Source data 2*.

• Source data 2. Annotated gels. Compilation of the annotated gels used for quantification of RNP and HDR integration efficiencies.

• Source data 3. Numeric data. Compilation of the numeric data used throughout the manuscript.

### Data availability
All data generated during this study are included in the manuscript and supporting files.

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
