## [Editor Report · eLife assessment]

This study presents an **important** methodology to increase the efficiency and precision of gene editing in human hematopoietic stem and progenitor cells. The evidence supporting the claims is **convincing** in that primitive LTC-ICs were minimally affected as a result of the editing procedure and the lack of edits at predicted off-target sites. The work will be of interest to biologists studying hematopoietic stem and progenitor cells and genome editing for potential clinical applications.

---

## [Referee Report · Reviewer #2 (Public review)]

Summary:

This work by Cloarec-Ung et al. sets out to uncover strategies that would allow for the efficient and precision editing of primitive human hematopoietic stem and progenitor cells (HSPCs). Such effective editing of HSPCs via homology directed repair has implications for the development of tractable gene therapy approaches for monogenic hematopoietic disorders as well as precise engineering of these cells for clinical regenerative and/or cell therapy strategies. In the setting of experimental hematology, precision introduction of disease relevant mutations would also open the door to more robust disease modeling approaches. It has been recognized that to encourage HDR, NHEJ as the dominant mode of repair in quiescent HSPCs must be inhibited. Testing editing of human cord blood HSPCs the authors first incorporate a prestimulation phase then identify optimal RNP amounts and donor types/amounts using standard editing culture conditions identifying optimal concentrations of AAV and short single-stranded oligonucleotide donors (ssODNs) that yield minimal impacts to cell viability while still enabling heightened integration efficiency. They then demonstrate the superiority of AZD7648, an inhibitor of NHEJ-promoting DNA-PK, in allowing for much increased HDR with toxicities imparted by this compound reduced substantially by siRNAs against p53 (mean targeting efficiencies at 57 and 80% for two different loci). Although AAV offered the highest HDR frequencies, differing from ssODN by a factor by ~2-fold, the authors show that spacer breaking sequence mutations introduced into the ssODN to better mimic the disruption of the spacer sequence provided by the synthetic intron in the AAV backbone yielded ssODN HDR frequencies equal to that attained by AAV. By examining editing efficiency across specific immunophenotypically identified subpopulations they further suggest that editing efficiency with their improved strategy is consistent across stem and early progenitors and use colony assays to quantify an approximate 4-fold drop in total colony numbers but no skewing in the potentiality of progenitors in the edited HSPC pool. Finally, the authors provide a strategy using mutation-introducing AAV mixed with different ratios of silent ssODN repair templates to enable tuning of zygosity in edited CD34+ cells.

Strengths:

The methods are clearly described and the experiments for the most part also appropriately powered. In addition to using state-of-the-art approaches, the authors also provided useful insights into optimizing the practicalities of the experimental procedures that will aid bench scientists in effectively carrying out these editing approaches, for example avoiding longer handling times inherent when scaling up to editing over multiple conditions.

The sum of the adjustments to the editing procedure have yielded important advances towards minimizing editing toxicity while maximizing editing efficiency in HSPCs. In particular, the significant increase in HDR facilitated by the authors' described application of AZD7648 and the preservation of a pool of targeted progenitors is encouraging that functionally valuable cell types can be effectively edited.

The discovery of the effectiveness of spacer breaking changes in ssODNs allowing for substantially increased targeting efficiency is a promising advance towards democratizing these editing strategies given the ease of designing and synthesizing ssODNs relative to the production of viral donors.

The ability to zygosity tune was convincingly presented and provides a valuable strategy to modify this HDR procedure towards more accurate disease modelling.

Weaknesses:

Despite providing convincing evidence that functional progenitors can be successfully edited by their procedure, as the authors acknowledge it remains to be verified to what degree the survival/self-renewal capacity and in vivo regenerative potential of the more primitive fractions is maintained with their strategy. That said the inclusion of LTC-IC assays that verify the lack of effect on these quite primitive cells is encouraging that functionality of stem cells will be similarly spared.

---

## [Author Response]

The following is the authors’ response to the previous reviews.

**Public Reviews**

**Reviewer #1 (Public Review):**
Summary:The findings in this manuscript are important in the gene editing in human-derived hematopoietic stem and progenitor cells. By optimizing the delivery tool, adding DNA-PK inhibitor and including spacer-breaking silent mutations, the editing efficiency is significantly increased, and the heterozygosity could be tuned. The editing is even across the hematopoietic hierarchy.Strengths:The precise gene editing is important in gene therapy in vitro and in vivo. The manuscript provides solid evidence showing the efficacy and uniqueness of their gene editing approach.Weaknesses:There are several extended and unique points shown in this paper but in a specific cell population.

The findings are indeed in a specific cell lineage, though it should be noted the editing crossed multiple cell types within that lineage. More importantly though, HSPC have substantial relevance to understanding adult stem cell biology, blood formation, and leukemia. Critically, they are also the target cells for a plethora of gene therapies for anemias, immunodeficiencies, metabolic disorders, and are also being explored for use with CAR technologies. Indeed, CRISPR-based gene therapy was recently approved for clinical use. As such, the findings here are of substantial relevance for multiple areas of research including hematology, stem cell biology, cancer, immunology and more.

**Reviewer #2 (Public Review):**
Summary:This work by Cloarec-Ung et al. sets out to uncover strategies that would allow for the efficient and precision editing of primitive human hematopoietic stem and progenitor cells (HSPCs). Such effective editing of HSPCs via homology directed repair has implications for the development of tractable gene therapy approaches for monogenic hematopoietic disorders as well as precise engineering of these cells for clinical regenerative and/or cell therapy strategies. In the setting of experimental hematology, precision introduction of disease relevant mutations would also open the door to more robust disease modeling approaches. It has been recognized that to encourage HDR, NHEJ as the dominant mode of repair in quiescent HSPCs must be inhibited. Testing editing of human cord blood HSPCs the authors first incorporate a prestimulation phase then identify optimal RNP amounts and donor types/amounts using standard editing culture conditions identifying optimal concentrations of AAV and short single-stranded oligonucleode donors (ssODNs) that yield minimal impacts to cell viability while still enabling heightened integration efficiency. They then demonstrate the superiority of AZD7648, an inhibitor of NHEJ-promoting DNA-PK, in allowing for much increased HDR with toxicities imparted by this compound reduced substantially by siRNAs against p53 (mean targeting efficiencies at 57 and 80% for two different loci). Although AAV offered the highest HDR frequencies, differing from ssODN by a factor by ~2-fold, the authors show that spacer breaking sequence mutations introduced into the ssODN to better mimic the disruption of the spacer sequence provided by the synthetic intron in the AAV backbone yielded ssODN HDR frequencies equal to that attained by AAV. By examining editing efficiency across specific immunophenotypically identified subpopulations they further suggest that editing efficiency with their improved strategy is consistent across stem and early progenitors and use colony assays to quantify an approximate 4-fold drop in total colony numbers but no skewing in the potentiality of progenitors in the edited HSPC pool. Finally, the authors provide a strategy using mutation-introducing AAV mixed with different ratios of silent ssODN repair templates to enable tuning of zygosity in edited CD34+ cells.Strengths:The methods are clearly described and the experiments for the most part also appropriately powered. In addition to using state of the art approaches the authors also provided useful insights into optimizing the practicalities of the experimental procedures that will aid bench scientists in effectively carrying out these editing approaches, for example avoiding longer handling times inherent when scaling up to editing over multiple conditions.The sum of the adjustments to the editing procedure have yielded important advances towards minimizing editing toxicity while maximizing editing efficiency in HSPCs. In particular, the significant increase in HDR facilitated by the authors' described application of AZD7648 and the preservation of a pool of targeted progenitors is encouraging that functionally valuable cell types can be effectively edited.The discovery of the effectiveness of spacer breaking changes in ssODNs allowing for substantially increased targeting efficiency is a promising advance towards democratizing these editing strategies given the ease of designing and synthesizing ssODNs relative to the production of viral donors.The ability to zygosity tune was convincingly presented and provides a valuable strategy to modify this HDR procedure towards more accurate disease modelling.Weaknesses:Despite providing convincing evidence that functional progenitors can be successfully edited by their procedure, as the authors acknowledge it remains to be verified to what degree the self-renewal capacity and in vivo regenerative potential of the more primitive fractions is maintained with their strategy.

As other the 53BP1-based editing strategy that also disrupt DNA-PK have demonstrated maintained allele frequencies over engraftment time (De Ravin et al. Blood 2021), this suggests that a transient disruption of DNA-PK shouldn’t compromise regenerative potential. Of course, we strongly agree that maintained regenerative potential is important in any editing strategy. As such, for the version of record we have added clonal LT-CIC assessment using conditions that we’ve previously demonstrated predict long-term repopulating potential (Knapp et al. Nat Cell Bio 2018). This data, which has been added to Figure 3, shows no significant reduction in the frequency of the most potent LT-CIC in edited cells compared to unedited controls.

Assessments of the potential for off-target effects via the authors' approach was somewhat cursory and would have benefited from a more thorough evaluation.

Once again in the 53BP1 strategy, the authors of that study already performed CHANGE-seq, long-range PCR, NGS, and SKY with inhibition of this same pathway without obvious increases in off-target editing (as long as HDR donor was present, though they did interestingly observe increased large deletions when HDR donors were absent, De Ravin et al. Blood 2021). Our tests here were designed to confirm that our molecule was similarly not affecting off-target editing rather than to launch a large-scale investigation. We agree, however, that off-targets and particularly structural re-arrangements that could be missed by other approaches remain a concern. We have added in nanopore sequencing of the predicted off-target sites and thus verified more deeply that there was no change (indeed no observable off-target activity) at any of these sites. This data has been added to Figure 2 and to a new supplementary Figure, Figure 2-figure supplement 3. Additionally, while it’s beyond the scope of the current manuscript, a focused follow-up dedicated to structural rearrangements downstream of both single and multiple edits is currently in progress and will be submitted separately later this year.

Viability was assessed by live cell counting however given the short-term nature of the editing assay, more sensitive readouts of potentially compromised cell health could have provided a more stringent assessment of how the editing methodology impacted cell fitness.

Of course, we agree that viable cell counting does not fully predict whether the cell is viable in terms of retained proliferative potential or other functional potentials. This point was addressed for myeloid progenitors at least by the CFC assays already in the manuscript, as to form a colony these cells were definitionally viable at input. Indeed, in these tests, we did see a reduction beyond that of the viable counts as already discussed in the text. Similarly, we already inadvertently answered this in the general CD34+CD45RA- population in Figure 4C where we measured clonal growth following editing with different mutant to silent donor ratios. In this instance we observed 30-40% clonogenic frequencies (Figure 4C), though in this case without a specific non-edited control (as this was not the intended question). None-the-less, this would indicate that any general viability loss was no more than observed in the CFC tests (even if we assume 100% cloning efficiency if the cells had been unedited). Finally, the clonal LTC-IC show that while there is perhaps some loss in more committed progenitors, those with the highest self-renewal potential are not compromised in the edited condition compared to control (Figure3I).

**Recommendations for the authors**

**Reviewer #2 (Recommendations For The Authors):**
It will be important to include the author-provided new paragraph in the discussion to contextualize this work in the existing HSPC editing landscape and your unique findings.

A new paragraph detailing how our manuscript fits with other recently published works is now included in the discussion.

The legend for Figure 3 needs correction. Panel E is incorrectly labeled as panel D and panel F is incorrectly labeled as panel E.

Thank you for catching this typo. It has been fixed.

In Figure 4 axis headings in panel C and D require clarity beyond simply titles of "Mean Frequency".

These axis labels have been clarified.